# The pseudoknot region and poly-(C) tract comprise an essential RNA packaging signal for assembly of foot-and-mouth disease virus

**Chris Neil** [1]*, **Joseph Newman**[1], **Nicola J. Stonehouse**[2], **David J. Rowlands**[2], **Graham J. Belsham**[3,4], **Tobias J. Tuthill** [1]*

**1** The Pirbright Institute, Ash Road, Pirbright, Surrey, United Kingdom, **2** The Faculty of Biological Sciences, University of Leeds, Leeds, United Kingdom, **3** Department of Veterinary and Animal Sciences, University of Copenhagen, Copenhagen, Denmark, **4** DTU National Veterinary Institute, Technical University of Denmark, Lindholm, Kalvehave, Denmark

* chris.neil@pirbright.ac.uk (CN); toby.tuthill@pirbright.ac.uk (TJT)

## Abstract

Virus assembly is a crucial step for the completion of the viral replication cycle. In addition to ensuring efficient incorporation of viral genomes into nascent virions, high specificity is required to prevent incorporation of host nucleic acids. For picornaviruses, including FMDV, the mechanisms required to fulfil these requirements are not well understood. However, recent evidence has suggested that specific RNA sequences dispersed throughout picornavirus genomes are involved in packaging. Here, we have shown that such sequences are essential for FMDV RNA packaging and have demonstrated roles for both the pseudoknot (PK) region and the poly-(C) tract in this process, where the length of the poly-(C) tract was found to influence the efficiency of RNA encapsidation. Sub-genomic replicons containing longer poly-(C) tracts were packaged with greater efficiency *in trans*, and viruses recovered from transcripts containing short poly-(C) tracts were found to have greatly extended poly-(C) tracts after only a single passage in cells, suggesting that maintaining a long poly-(C) tract provides a selective advantage. We also demonstrated a critical role for a packaging signal (PS) located in the pseudoknot (PK) region, adjacent to the poly-(C) tract, as well as several other non-essential but beneficial PSs elsewhere in the genome. Collectively, these PSs greatly enhanced encapsidation efficiency, with the poly-(C) tract possibly facilitating nearby PSs to adopt the correct conformation. Using these data, we have proposed a model where interactions with capsid precursors control a transition between two RNA conformations, directing the fate of nascent genomes to either be packaged or alternatively to act as templates for replication and/or for protein translation.

## Author summary

Genome packaging, whereby viral RNA is incorporated into protective protein capsids to produce more virus particles, is a crucial step in RNA virus life cycles. It is a stringent process as only viral RNA is encapsidated, while cellular RNA is excluded.

**Data Availability Statement:** All data are in the manuscript and/or supporting information files.

**Funding:** This work was funded by BBSRC (BB/V008323/1), TJT, and internal funding from The Pirbright Institute (BBS/E/I/00007034 and BBS/E/PI/230002A), the National Veterinary Institute at the Technical University of Denmark (DTU) and the University of Copenhagen, GJB. Experimental work was carried out in the High Containment Facility at The Pirbright Institute (BBS/E/I/00007037 and BBS/E/PI/23NB0004). The authors would like to acknowledge the Bioinformatics, Sequencing and Proteomics unit and support through the Core capability grant (BBS/E/I/00007039). At DTU and UCPH, work was funded by the Danish Veterinary and Food Administration (FVST) as part of the agreement for commissioned work between the Danish Ministry of Food and Agriculture and Fisheries and DTU and then with the University of Copenhagen. NJS and DJR received funding from BBSRC (BB/K003801/1 and BB/T015748/1). The funders had no role in study design, data collection and analysis, decision to publish, or preparation of the manuscript.

**Competing interests:** The authors have declared that no competing interests exist.

This study reveals the essential role of packaging signals in FMDV RNA encapsidation, specifically those in the pseudoknot region and in a region that can contain >100 cytosines, termed the poly-(C) tract. We demonstrate that the length of the poly-(C) tract significantly affects packaging efficiency; genomes containing longer poly-(C) tracts are favoured. This is the first role that has been identified for the poly-(C) tract in FMDV. We have also found an essential packaging signal in the pseudoknot region, which is assisted by other packaging signals located throughout the genome, that together facilitate encapsidation of FMDV RNA. These results provide compelling evidence for the involvement of RNA packaging signals in FMDV assembly. Based on this, we propose a simple model for FMDV RNA packaging, which involves a transition from genome replication to genome packaging and is controlled by packaging signals. This knowledge could pave the way for future research and development of novel antiviral strategies targeting FMDV and other picornaviruses.

## Introduction

Foot-and-mouth disease virus (FMDV), a species within the *Aphthovirus* genus of the family *Picornaviridae*, is the aetiological agent of foot-and-mouth disease (FMD) in cloven-hooved animals [1]. FMD is extremely contagious and is endemic in parts of Africa and Asia. It is controlled through vaccination in regions at risk of infection, while countries including the USA, UK, Australia and much of mainland Europe maintain FMD-free status without vaccination. However, outbreaks of the disease in countries that are normally FMD-free can cause severe economic losses and the ongoing cost to combat the virus in endemic countries is significant [2].

FMDV is a small, non-enveloped, single-stranded, positive-sense RNA virus with a genome approximately 8.4 kb in length that is packaged into a pseudo $T = 3$ icosahedral capsid [3]. The genome contains a single large open reading frame, encoding both the structural and non-structural proteins. The first protein in this polyprotein is the Leader protease, followed by the capsid precursor P1-2A, and the non-structural proteins 2B, 2C, 3A, $3B_{(1-3)}$, $3C^{pro}$ and 3D [4]. The capsid precursor P1-2A is cleaved by the viral protease $3C^{pro}$ into VP0, VP1, VP3 and 2A, of which VP0, VP1 and VP3 remain associated in the protomer [5]. Five protomers then assemble into a pentamer and 12 pentamers associate with a molecule of genomic RNA to form the complete infectious virus particle. Finally, there is an assembly dependent maturation cleavage of VP0 into VP2 and VP4 which is more efficient in the presence of packaged RNA, but can still occur independently [6]. Although this maturation cleavage is common to most picornaviruses, it is not ubiquitous. For example, viruses in the *Parechovirus* genus do not undergo this process and VP0 remains intact in the mature virus [7]. In addition to the formation of mature virions, assembly of capsids lacking RNA can also occur to produce non-infectious empty particles [8].

The FMDV genome is organised into a highly structured 5'-UTR, with a virus-encoded peptide (VPg) covalently bound to the 5' terminus of the RNA [9,10], the open reading frame; and a 3'-UTR terminating in a poly-A tail. The key elements of the 5'-UTR are the: S-fragment, poly-(C) tract, pseudoknot (PK) region, the *cis*-acting replication element (*cre*) and the internal ribosome entry site (IRES). Of particular interest for this study are the poly-(C) tract and the adjacent PK region, neither of which are present in some more intensely studied picornaviruses, e.g. enteroviruses.

The poly-(C) tract typically consists of a stretch of about 50–200 cytosines [11]. The requirement for the poly-(C) tract to be over a certain length has been linked to virulence in both cardioviruses and aphthoviruses [12,13]. However, conflicting results have also been reported [14]. This is, in part, due to the difficulty of cloning such a long homopolymer and a mechanism allowing truncated poly-(C) tracts to revert to a *wt* length during passage, thus obscuring potential differences [15,16]. However, it has been shown that this feature is not essential for replication.

The PK region is made up of 2–4 PKs with high sequence similarity, which directly follow the poly-(C) tract. While at least one PK must be present for *wt* levels of replication, RNA transcripts lacking all four PKs are still able to replicate. Despite this, we have shown that RNA transcripts without the PKs cannot be recovered as infectious virus, which we hypothesised may be due to defects in packaging [17].

Picornavirus RNA is thought to be encapsidated during capsid assembly and, although no clear mechanism has been established for how this is achieved, recent work has highlighted the role of RNA packaging signals (PSs) in this process. PSs are used by other RNA viruses to incorporate viral RNA into capsids, with the bacteriophage MS2 being seen as a model system for PS-mediated packaging [18]. Additionally, PSs have been identified in other picornaviruses–most notably aichi virus, human parechovirus 1 and enterovirus-E [19–25]. In these systems, multiple small, low affinity PSs are dispersed across the viral genome, which function in concert to package the genome more efficiently into the virus capsid [26]. These PSs may each recruit a capsid subunit, for example a picornavirus pentamer, sequentially in Hamiltonian pathways [27]. The arrangement of PSs throughout the genome, each with different affinities for the capsid subunits, may coordinate the assembly process by initially binding to PSs with the strongest affinities, followed by recruitment of the lower-affinity PSs to complete the capsid structure. For some viruses, the presence of PSs is essential for capsid formation, for example MS2 PS-coat protein interactions are required for a conformational change in the coat protein which enables efficient assembly [28]. Although FMDV pentamers can assemble into empty capsids in the absence of RNA (and PSs are therefore not essential for capsid formation) these PSs may facilitate and stabilise the process, making assembly more efficient and preventing the capsid from dissociating prematurely. Additionally, other studies on picornaviruses have described alternative mechanisms for encapsidation. For the enterovirus poliovirus (PV), for example, all non-essential sections of the genome have been either deleted or extensively mutated while maintaining the encoded protein sequence, or swapped with those of other picornaviruses without affecting virus viability, suggesting that PV RNA encapsidation does not rely solely on PSs [29–32]. As an alternative to RNA PSs, an interaction between 2C, a viral protein involved in RNA replication, and the capsid protein VP3 was proposed as the link between replication and encapsidation [33]. However, recent work has identified PS motifs across the PV genome [25], which suggests that picornaviruses may employ both protein-protein interactions and RNA-protein interactions during encapsidation.

FMDV packaging appears to be extremely stringent since only genomic RNA (or FMDV replicon RNA) is packaged into virus capsids [34]. Additionally, we have previously identified functional PSs dispersed across the FMDV genome [35]. The 5'-most PS identified is located in the PK region, and deletion of sequences within this region prevented virus recovery [17].

A number of factors influence the assembly of infectious virus, including the relative rates of protein and RNA synthesis, and these complicate the study of genome encapsidation in picornaviruses generally and in FMDV specifically. This is particularly relevant when the final assay of virus assembly is the measurement of virus viability. Approaches to identify picornavirus packaging signals have helped to establish the role of RNA-capsid interactions during virus assembly [24,25,35], but even these are limited by the reliance on virus viability for validation

and more direct methods to assess assembly and encapsidation are needed. To this end, an assay designed to look at the relative encapsidation efficiency of GFP-expressing replicons *in trans* was developed based on the assays carried out by Barclay et al. and McInerney et al. [30,34]. This assay demonstrated that poliovirus replicons could be packaged into poliovirus capsids *in trans*, but the assay was significantly less efficient when FMDV was used.

In the work presented here, we have further optimised the *trans*-encapsidation assay to enable the study of FMDV RNA packaging, solving the problem of inefficient replicon *trans*-encapsidation. We then used this assay to demonstrate the importance of both the poly-(C) tract and the neighbouring PSs in the PK region with regards to RNA packaging and propose a model for FMDV packaging that relies on PSs dispersed across the genome.

## Methods

### Cell lines

The baby hamster kidney-21 (BHK-21) cell line was obtained from the central services unit (CSU) at The Pirbright Institute. This cell line was used for all cell culture experiments. The cells were grown in Glasgow's minimum essential media (GMEM, Merck) with 10% foetal bovine serum (FBS; Gibco), 2 mM L-glutamine (Merck), 100 units/L penicillin, 100 μg/mL streptomycin (Merck) and 5% tryptose phosphate broth (Gibco). For virus infections, virus growth media (VGM) was used, which contained the same components but with either 1% or 5% FBS.

Cells used for transfection and/or infection were seeded the previous day using a suitable dilution to reach 80% confluence at the start of the experiment.

### Plasmids

Three plasmids were used as the basis for the work described here: an O1 Kaufbeuren (O1K) FMDV ΔP1 ptGFP replicon, containing the Leader protease coding region but with a ptGFP coding region replacing the capsid coding region [36]; an O1K aqGFP Leaderless (ΔLb) infectious copy plasmid (ΔLb.GFP.FMDV), which does not contain the Leader protein coding region but does contain the aqGFP coding region [37]; and the full-length infectious copy of O1K FMDV, pT7S3 [38]. These are shown in Fig 1.

Four variants of the ΔP1 GFP replicon were used here: C11 ΔPK1234, ΔPK234, ΔPK34 and C11, which we have previously described [17]. An additional variant, C29, was made for this study by ligating synthetic DNA from GeneArt containing the alternative-length poly-(C) tract.

The ΔLb.GFP.FMDV infectious copy plasmid was modified to make a replicon by changing the encoded 3Cpro cleavage sites to prevent P1-2A processing. This replicon was called the Leaderless defective-capsid GFP replicon (ΔLbdcap GFP replicon). The sequence encoding the VP2/VP3 junction was changed from GAGGGA to GCTGCT (PSKE/GIFP to PSKA/AIFP), the sequence encoding the VP3/VP1 junction was changed from GAAACC to GCCGCA (ARAE/TTSA to ARAA/ATSA) and a codon in the VP0 myristoylation site sequence was changed from GGG to GCC (VP0 G1A) by the introduction of a synthetic fragment of DNA from GeneArt. Variations were made to the ΔLbdcap GFP replicon, again by incorporating synthetic DNA from GeneArt, to make the: C11, C29, C39 ΔPK1 and C40 ΔPK2 ΔLbdcap GFP replicons.

The pT7S3 plasmids containing truncated poly-(C) tracts were made by digesting the corresponding ΔLbdcap GFP replicons with suitable restriction enzymes and ligating in the fragment containing the poly-(C) tract to replace the equivalent region of pT7S3.

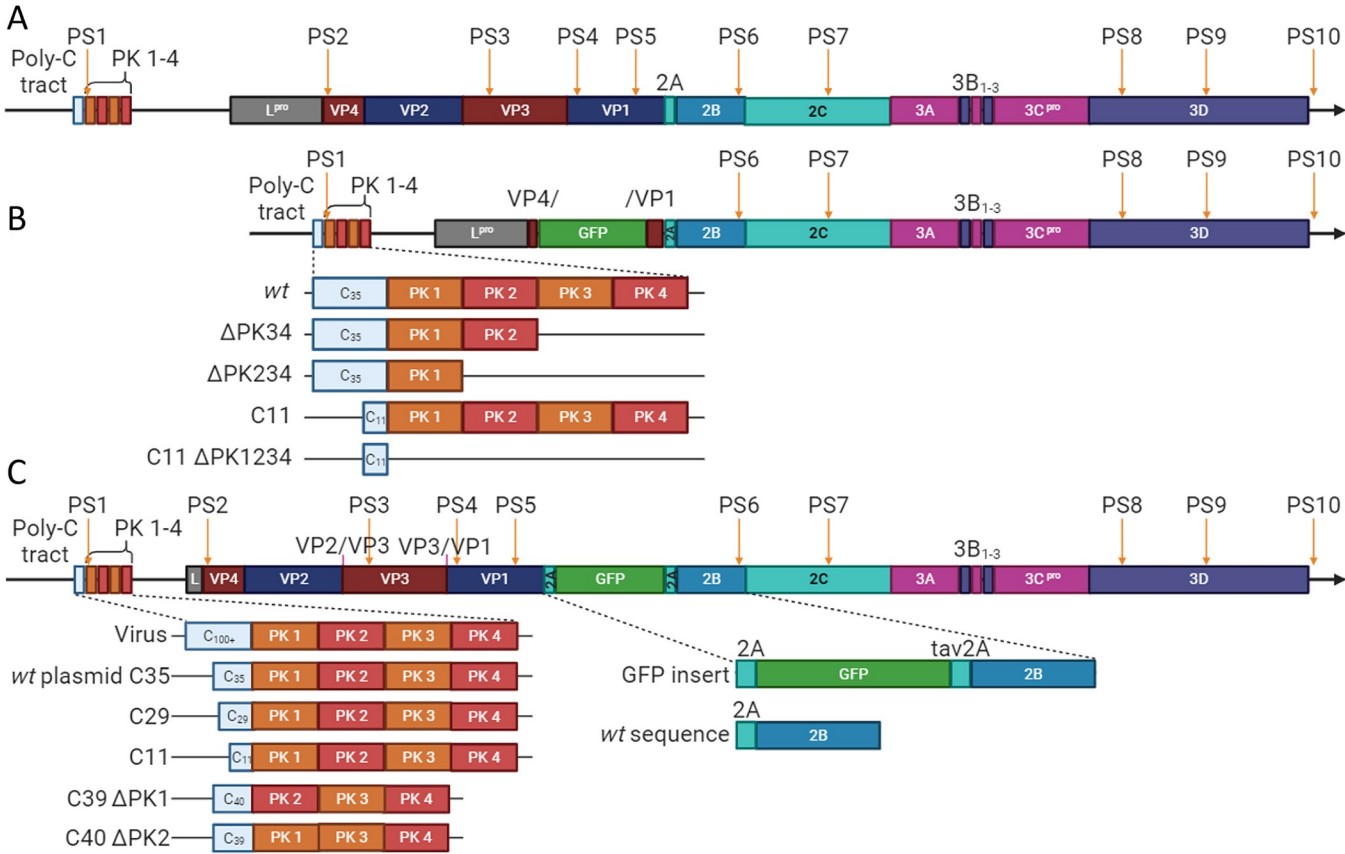

**Fig 1. Schematic of selected features within the virus, ΔP1 GFP replicon and ΔLbdcap GFP replicon.** Representations of the (A) pT7S3 viral genome, (B) ΔP1 GFP replicon (including the ΔPK34, ΔPK234, C11 and C11 ΔPK1234 versions) and (C) ΔLbdcap GFP replicon (including the *wt* C35, C29, C11, C39 ΔPK1 and C40 ΔPK2 versions). Substitutions are labelled above their respective positions (VP2/VP3 and VP3/1 3C^pro cleavage sites), and packaging signals (described by Logan et al. [35]) are labelled above with orange arrows.

All restriction enzyme digests were performed using enzymes from New England Biolabs (NEB) based on the product protocols. The vector portions were dephosphorylated using rAPid alkaline phosphatase (Roche) following digestion with the appropriate restriction enzymes. Fragments were then separated using DNA electrophoresis, purified using the Monarch DNA Gel Extraction kit (NEB) and ligated using T4 DNA ligase (Invitrogen).

### *In vitro* transcription

Plasmids were linearised using appropriate restriction enzymes (*AscI* for the ΔP1 ptGFP based replicons, *HpaI* for all others). The DNA was then purified using the Monarch PCR & DNA Cleanup Kit (NEB), and RNA was synthesised from the purified DNA using the MEGAscript T7 Transcription Kit (Invitrogen) according to the manufacturer guidelines. Input DNA was digested using the Turbo included in the kit. The RNA was purified using a MEGAclear kit (Invitrogen) and quantified using the Qubit RNA BR Assay Kit (Invitrogen) with a Qubit Fluorometer 4 according to the manufacturer's protocol.

### BHK cell transfections

BHK cells were transfected with RNA using Lipofectamine 2000, based on the manufacturer's protocol (Invitrogen). The RNA was mixed with Opti-Mem (Gibco), with 1 µg (suitable for a

24-well transfection) made up to 50 μL. The Lipofectamine 2000 was diluted 1 in 10 in Opti-Mem to make an equal volume of Lipofectamine 2000-Opti-Mem mix (per RNA sample). Each was incubated at room temperature for 5 minutes, before the RNA/Opti-Mem solution was mixed with an equal volume of the Lipofectamine 2000/Opti-Mem solution and incubated for a further 15 minutes at room temperature. The medium was aspirated from cell monolayers and replaced with the RNA transfection mix, along with sufficient Opti-Mem to cover the cells and prevent them from drying out. The medium was replaced after two hours with an appropriate volume of 1% VGM. When a replication inhibitor was used with transfected replicons, this was achieved by adding guanidine hydrochloride (GuHCl) to both the transfection medium and the VGM at a final concentration of 3 mM.

### *Trans*-encapsidation assay

In the first round of the assay, BHK-21 cells were seeded in multi-well plates (6-, 12- or 24-well, Scientific Laboratory Supplies) and transfected with an appropriate amount of the replicon RNA transcript (4, 2 and 1 μg respectively). If an infectious clone transcript such as the C11 transcript was used as the helper genome, it was co-transfected with the replicon RNA by including it with the RNA-Opti-Mem mixture (no change to final volume or amount of Lipofectamine 2000 used). If a helper virus was used, it was added to the cells at 1-hour post-transfection. At 2-hours post-transfection the medium was replaced with an appropriate volume of VGM containing 1% serum to cover the cells. Cells transfected with the ΔP1 GFP replicon were incubated until 5 hours post-transfection, while cells transfected with the ΔLbdcap GFP replicon were incubated until 7 hours post-transfection, both at 37°C, before they were frozen at -20°C overnight. Duplicate plates were prepared in parallel and were analysed using the Incucyte S3 Live-Cell Analysis System, which imaged the cells hourly using phase light and detected fluorescence (emission wavelength 524 nm, excitation wavelength 460 nm). Images were analysed to calculate the number of GFP expressing cells present and the mean fluorescence intensity (MFI) was determined as a measure of replicon replication.

For the second round of the assay, the cells were thawed and pelleted, and the supernatant, containing encapsidated RNA, was treated with 1 μL (10 μg) RNase A (ThermoFisher Scientific) per 100 μL of supernatant for 10 minutes at room temperature to remove input transcripts. Fresh cells were then inoculated with 50, 100 or 200 μL of supernatant (24-, 12- and 6-well plates respectively) and loaded into the Incucyte S3 for hourly imaging using phase light and fluorescence detection (as above). Several images were taken per well which were analysed in real time for the proportion of the well covered with cells (% confluence), the number of fluorescent objects per well (GFP objects per well) and/or the MFI of the well where relevant. The parameters of the analysis were adjusted based on the characteristics of genuine GFP-expressing cells to optimise detection of GFP-expressing cells and to minimise background auto-fluorescence, which included setting minimum thresholds for both the intensity of the GFP and the area of the foci. Typical thresholds included: a threshold of 1.0 GCU (green calibrated unit); a maximum area of 1000 $\mu m^2$; a maximum eccentricity of 0.7; and a minimum integrated intensity (GCU x $\mu m^2$) of 2000. All other parameters were typically left at the default setting. Plates were typically read for 24 hours or until the GFP object count started decreasing.

### CPE development assay

The CPE (cytopathic effect) development assay was performed by transfecting a 96-well plate of BHK-21 cells with RNA transcripts made from the virus infectious clone plasmids. Plates were loaded into the Incucyte S3 Live-Cell Analysis System and imaged using phase light every

four hours until either complete CPE was reached, or the cell only negative control became unhealthy. Multiple images were taken per well, which were analysed in real time to determine the proportion of the well covered in cells to calculate the confluence of the cell monolayers. The parameters of the analysis were adjusted to optimise detection of healthy cells compared to unhealthy cells to better observe the spread of the rescued virus through the culture.

### RNA extraction and digestion

RNA transcripts were transfected into BHK21 cells, and cells were left to develop CPE. At the point of complete CPE, the cells were lysed by freeze/thawing and the lysate was clarified by centrifugation. A portion of the clarified lysate was then added to fresh BHK21 cells with VGM, and the cells were again left until complete CPE was reached. At this point, cells were processed using a Qiagen RNeasy Kit (74004) to extract the RNA. The purified RNA samples were then digested with RNase T1 (ThermoFisher Scientific) to release the poly-(C) tract; 2 μL (2000 U) of RNase T1 was added to the RNA and samples were incubated for 30 minutes at 37˚C.

### Automated electrophoresis

DNA and RNA samples were analysed using either the Agilent 4200 Tapestation (Tapestation D1000 High Sensitivity DNA kit and Tapestation RNA ScreenTape kit, Agilent) or the Agilent 2100 Bioanalyzer (DNA 12000 kit, Agilent). For the DNA kits, 2 μL of plasmid DNA (0.1–50 ng/μL) was added to 2 μL sample buffer, vortexed for 1 minute at 2000 rpm and loaded into the Tapestation/Bioanalyzer. For the RNA kit, 1 μL cellular RNA (25–200 ng/μL) was added to 5 μL sample buffer, heated to 72˚C in a thermocycler for 3 minutes, cooled to 4˚C for 2 minutes, vortexed for 1 minute at 2000 rpm and loaded into the Tapestation.

### RT-qPCR

Extracted RNA was analysed by RT-qPCR using the dye-based version of the Luna Universal One-Step RT-qPCR Kit (NEB) and primers against FMDV 3D (ACT GGG TTT TAC AAA CCT GTG A forward, GCG AGT CCT GCC ACG GA reverse). A total of 1 μL RNA was added to 19 μL mastermix and loaded into a thermocycler with the following conditions: 55˚C for 10 minutes, 95˚C for 1 minute and 45 cycles of 95˚C for 10 seconds followed by 60˚C for 30 seconds. The fluorophore was read based on the settings for FAM.

## Results

### A novel *trans*-encapsidation assay to investigate FMDV RNA packaging

The *trans*-encapsidation assay involves co-transfecting two RNA transcripts into cells: a GFP-expressing replicon which cannot produce a functional capsid and acts as the genome-donor; and an infectious copy transcript which acts as the capsid-donor. When these RNAs replicate in the same cell, the infectious copy produces functional capsid subunits, which can assemble together with either viral RNA derived from the infectious copy *in cis*, or with replicon RNA *in trans*. Progeny particles incorporating infectious genomes or replicon RNAs can then infect new cells, with cells infected by the *trans*-encapsidated replicon RNA being quantifiable by GFP expression. The relative t*rans*-encapsidation efficiencies of replicons containing mutations, which may affect encapsidation, can therefore be measured by comparing the GFP expression in monolayers of cells infected with lysates from the transfected cells of the first-round with that from a *trans*-encapsidated control replicon (Fig 2). The number of cells expressing GFP in the second round reflects the packaging efficiency of the *trans*-encapsidated

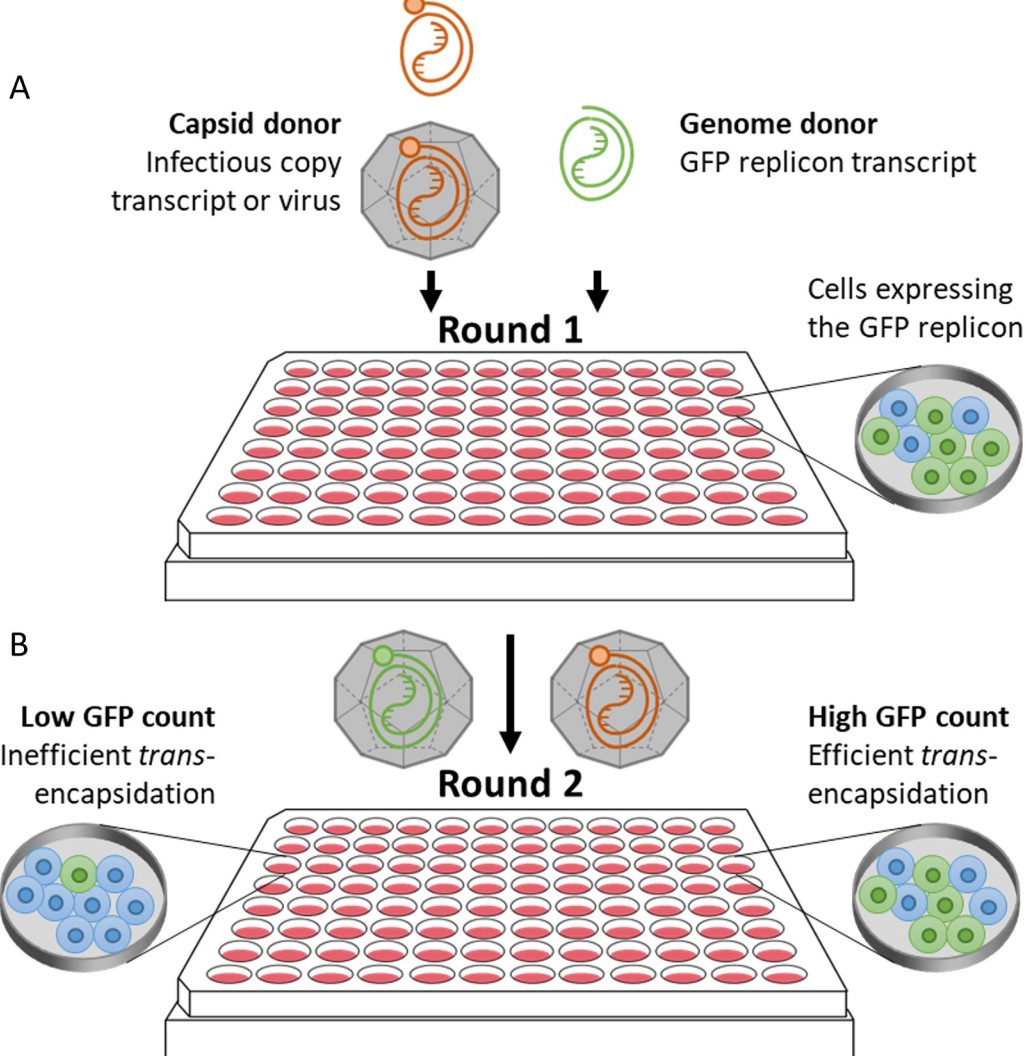

**Fig 2. *Trans*-encapsidation assay overview** (A) In the first round of the assay, cells were transfected with a GFP replicon transcript and either infected with *wt* virus or co-transfected with an infectious copy transcript, so the replicon and virus/ infectious copy transcripts were replicating in the same cells. This resulted in the production of both progeny *wt* virus and progeny trans-encapsidated virus particles, with the GFP replicon packaged into the capsid provided by the *wt* virus. The proportion of cells expressing GFP corresponds to the efficiency of transfection. (B) The cell lysate containing the progeny *wt* virus and trans-encapsidated virus particles was then used to infect a second round of fresh cells. Cells infected with the trans-encapsidated GFP replicon expressed GFP, and the proportion of GFP expressing cells compared to the standard GFP replicon sample represented the relative trans-encapsidation efficiency of mutant GFP replicons.

replicon. The transfection efficiencies and replication kinetics of the replicons used can be assessed from the GFP counts and mean fluorescent intensities (MFI) produced in the first round of the assay to ensure they are comparable. Where there were marked differences between the replicons, these were factored into the conclusions drawn from the experiments.

We hypothesised that the efficiency of viral RNA packaging would correlate with the number of PSs. To test this, two types of replicon RNA, with either normal or reduced numbers of PSs, were used in the *trans*-encapsidation assay: a GFP replicon with fewer PSs (due to replacement of the majority of the P1 capsid-coding region with the GFP reporter sequence, termed ΔP1 GFP replicon); and a Leaderless (ΔLb) GFP replicon including the capsid coding region

and the associated PSs, albeit with substitutions at the 3C$^{pro}$ cleavage sites to prevent capsid processing and assembly, termed the Leaderless defective capsid (ΔLbdcap) GFP replicon. Schematics of these constructs are shown in Fig 1.

Since the ΔP1 GFP replicon contains a GFP coding region in place of the capsid coding region, it is unable to produce infectious virions. Although this does not affect RNA replication and translation, the deletion of the capsid coding region also removes any PSs contained within this region, including four PSs identified by Logan et. *al.*: PS3, PS4, PS5 along with the majority of PS2 [35].

An alternative replicon was therefore designed with the maximum number of PSs possible, with the hypothesis that it would be packaged more efficiently in the *trans*-encapsidation assay. This alternative replicon retains the capsid coding region and the associated PSs. However, inclusion of both the capsid with the GFP reporter rendered the genome too large to be packaged [39]. Therefore, a ΔLb GFP virus was used where the deletion of the Lb portion of Leader coding region provides capacity in the genome for the GFP reporter. This region does not contain any currently identified PSs and has little effect on viability in BHK-21 cells if the remainder of Lab is still present [37]. Substitutions in the 3C$^{pro}$ cleavage sites within the capsid sequences were introduced to prevent the capsid proteins from being processed, thus preventing capsid assembly in the absence of helper viruses.

In the first round of the trans-encapsidation assay, the GFP signal from the ΔLbdcap GFP replicon was expressed less rapidly compared to the ΔP1 GFP replicon (Fig 3A). This indicates that the ΔLbdcap replicon replicates with slower kinetics, but it reached the same peak signal for both mean fluorescence intensity (Fig 3A) and GFP object count (which represents the number of GFP positive cells) (Fig 3B). GFP expression from the first round, both in terms of GFP object count and GFP MFI, was used to determine replicon replication fitness. GFP expression is dependent on both RNA replication ability and translation levels, so when GFP expression in the mutant constructs was comparable to the *wt* construct, it indicated that neither RNA replication nor translation were markedly impaired. Both samples were harvested at the point of peak GFP expression (5 and 7 hours respectively for the ΔP1 and ΔLbdcap GFP replicons) to ensure that these were harvested at equivalent points in the lifecycle. When treated with 3 mM guanidine hydrochloride (GuHCl), an RNA replication inhibitor, the GFP expression derived from translation of input RNA alone for the two replicons was comparable. Furthermore, it was lower than the GFP expression for the replicating replicons, indicating that GFP expression is dependent on both RNA replication and translation, and supporting our use of GFP expression as a method of monitoring changes in both processes (S1 Fig).

The second round of the *trans*-encapsidation assay revealed that the ΔLbdcap GFP replicon was packaged 25% more efficiently compared to the ΔP1 GFP replicon (Fig 3C). This indicated that the benefits for *trans*-encapsidation provided by including additional PSs outweigh the fitness costs incurred by deleting the Leader protein (Lb) coding sequence. When the first round was carried out in the presence of GuHCl, negligible levels of *trans*-encapsidation occurred (S1 Fig).

## The length of the poly-(C) tract affects encapsidation

The PK region and PS1 are located immediately adjacent to the poly-(C) tract (Fig 1), and for practical reasons the DNA fragments used to alter the PK region contained a synthetic poly-(C) tract which was shorter than the poly-(C) tract in the parental replicons. To investigate the possible influence of poly-(C) tract length on packaging, two variations of the ΔLbdcap GFP replicon were generated using synthetic DNA sequences containing poly-(C) tract lengths of 11 and 29 nt respectively for comparison to the *wt* ΔLbdcap replicon, which has approximately

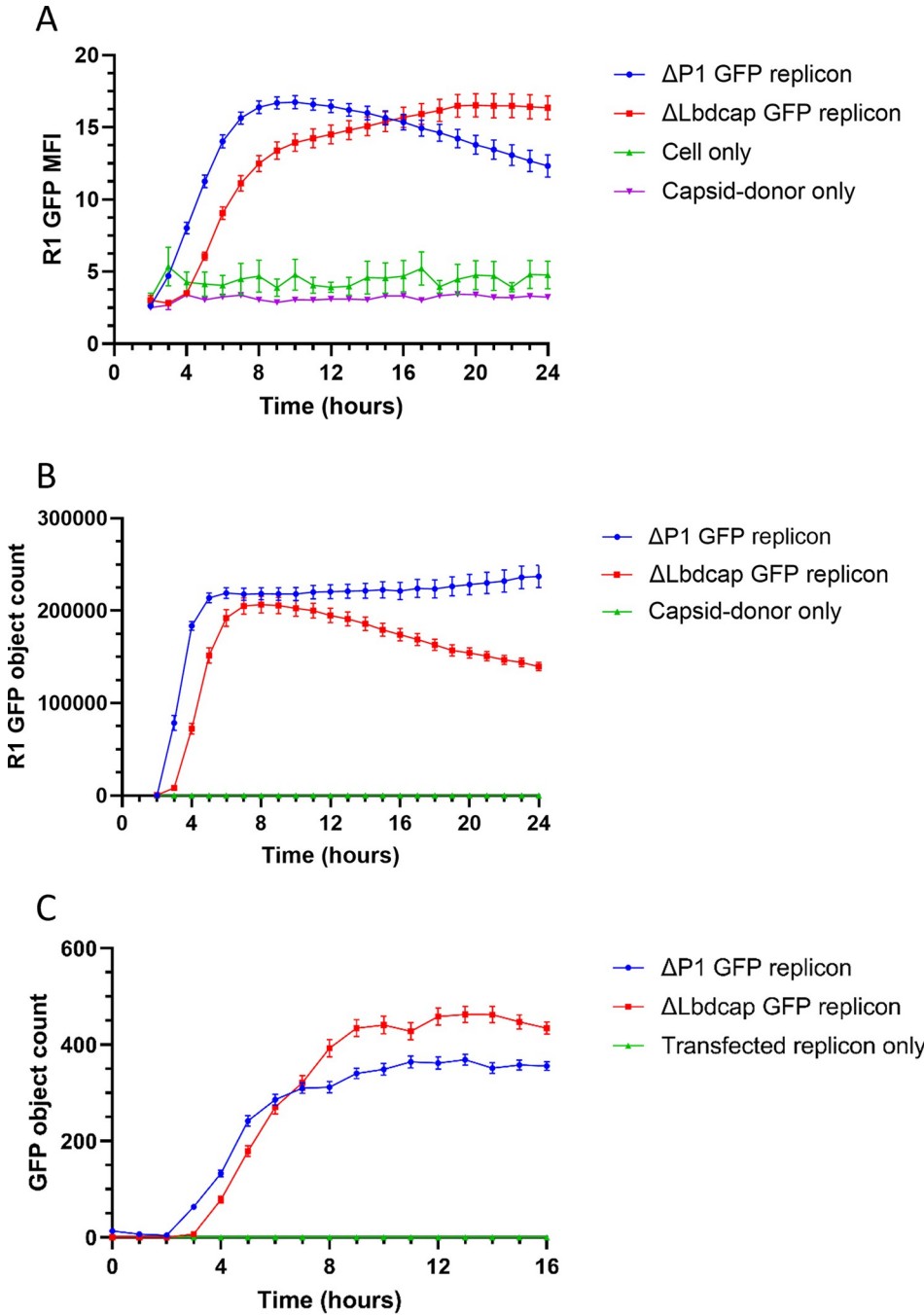

**Fig 3. The presence of packaging signals enhances *trans*-encapsidation efficiency.** Cells were transfected with the ΔP1 GFP replicon and ΔLbdcap GFP replicon, with and without virus co-infection, along with an uninfected ΔLbdcap GFP replicon transfection only control. (A) The MFI for the first round and the GFP counts for the (B) first and (C) second rounds of the trans-encapsidation assays are shown. 'GFP objects' are the GFP positive foci which are in the range of parameters defined in the analysis setup. Data shown are from a single experiment, representative of multiple experiments, and error bars represent the SEM calculated from (A-B) 5 and (C) 12 images.

35 cytosines. Furthermore, it should be noted that these poly-(C) tracts are all shorter than the ca. 50–200 nt present in the rescued viruses due to the difficulty of cloning and maintaining a long poly-(C) tract in the plasmid.

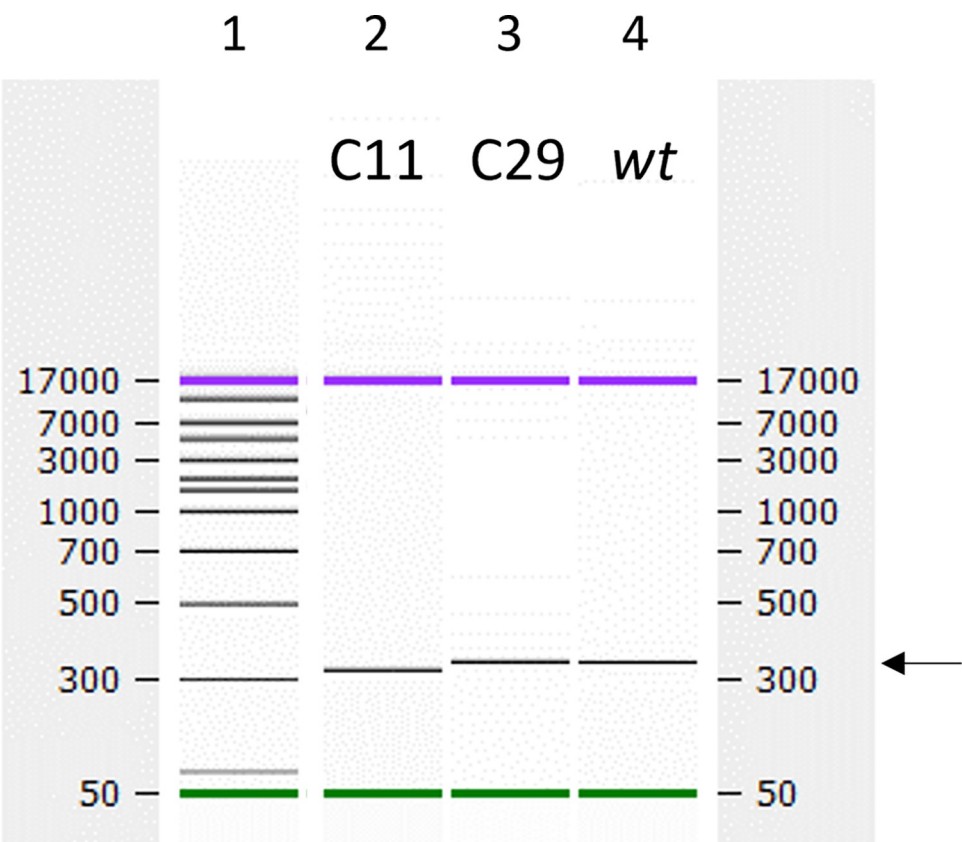

**Fig 4. The poly-(C) tract in the C11 ΔLbdcap GFP replicon is shorter than the poly-(C) tracts in the C29 and C35 ΔLbdcap GFP replicons.** Plasmids were digested using *Xba*I and *Nhe*I, gel purified to obtain the ca 340 bp fragment containing the poly-(C) tract and analysed using the Bioanalyser. The ladder (in bp) is shown in Lane 1, the C11 replicon in Lane 2, the C29 replicon in Lane 3 and the *wt* C35 replicon in Lane 4. The single analysis shown is representative of three experiments. The fragment of interest is indicated on the right by the arrow. The fragment from the C11 replicon was estimated to be 325–328 bp (C9-12), while the C29 and C35 *wt* replicon fragments were both estimated to be 347–352 bp (C30-36). Expected sizes were 327, 345 and 351 bp respectively.

The poly-(C) tract length of each construct was verified by digesting the plasmids at restriction sites to excise a fragment containing 316 base pairs (bp) plus the poly-(C) tract and analysing the size of each DNA fragment using electrophoresis (Fig 4). Although the resolution was not sufficient to determine the exact lengths of the poly-(C) tracts consistently across multiple experiments, it was possible to approximate the lengths in each analysis. The fragment sizes in each experiment were also compared to determine the relative differences in poly-(C) tract lengths between the three samples to confirm that the *wt* was largest in each experiment, as expected, followed by the C29 replicon and then the C11 replicon. The C11 poly-(C) tract was consistently shorter than that from both other replicons, and was estimated to contain between 9 and 12 cytosines. The C29 and *wt* replicons appeared to be of a similar size, with the *wt* replicon appearing either slightly larger or of the same size, consistent with the small difference in expected lengths. Both were estimated to be between 30 and 36 cytosines in length.

The RNA transcripts from these plasmids were then tested in the *trans*-encapsidation assay described above. No differences in replication of these replicons were observed in the first round, either by GFP counts or the MFI of the GFP objects (Fig 5A and 5B), indicating that the poly-(C) tract truncations had no measurable effect on replicon replication kinetics and therefore did not adversely affect either RNA replication or translation. In the second round,

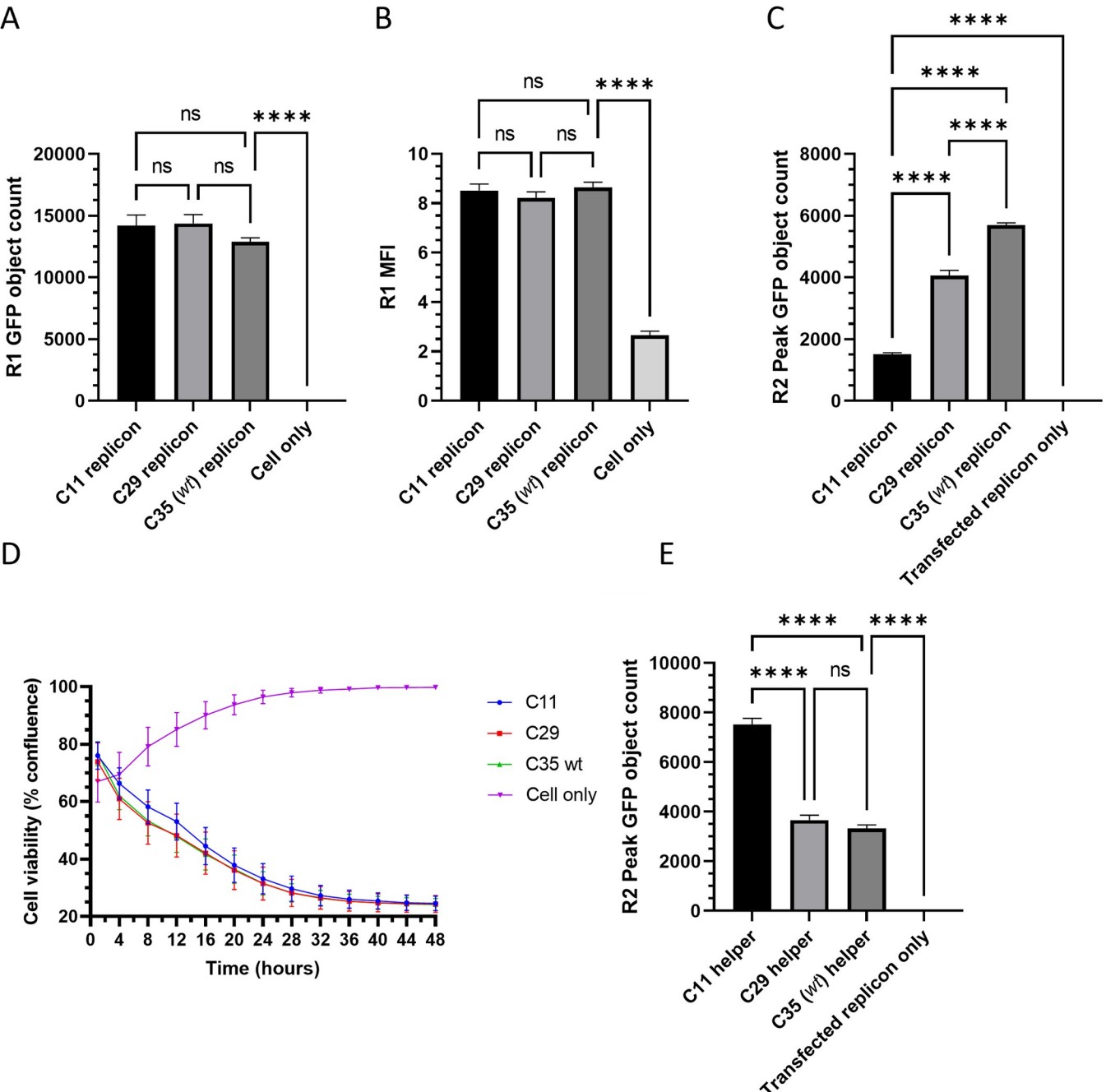

**Fig 5. Transcripts with shorter poly-(C) tracts are less competitive during encapsidation.** Replicons with differing length poly-(C) tracts were compared in the *trans*-encapsidation assay and were assessed according to the (A) first round GFP object count, (B) first round green MFI and (C) second round GFP object count. (D) RNA transcripts containing the equivalent poly-(C) tracts were compared in terms of speed of CPE development and (E) their ability to compete with a standard replicon for packaging resources, when used as the capsid-donor in the *trans*-encapsidation assay; high replicon *trans*-encapsidation efficiency equates to the capsid-donor being poorer at competing with the replicon. Data shown represent the mean from triplicate wells at either the point of harvest (A-B) or the time point with peak GFP expression (C and E), and error bars represent the SEM calculated from 5 (A-B), 12 (C and E), and 15 (D) images. Significance is shown for A-C and E comparing the samples to each other and between the *wt* replicon and the cell only/transfected replicon only controls using a one-way ANOVA (**** $p < 0.0001$). Significance in D was calculated using Wilcoxon tests between each sample, but none were significant ($p < 0.001$).

however, *trans*-encapsidation efficiency correlated with the length of the poly-(C) tract; the *wt* C35 replicon with the longest poly-(C) tract had the greatest peak GFP count of 5700 at 16 hours, while the C11 and C29 replicons had peak counts reaching only 1500 +/- 50 (26% of *wt*) and 4000 +/- 150 (71% of *wt*) respectively at 16 hours (Fig 5C). All differences were significant (p < 0.0001).

The effect of truncations to the poly-(C) tract on encapsidation efficiency has not been previously reported and no differences in virus viability have been observed, provided that the poly-(C) tract is at least 6 nucleotides in length [16]. We also saw no effect on RNA replication or ability to rescue virus from an infectious whole-genome RNA transcript with 11 cytosines in the poly-(C) tract [17]. To extend these observations, we investigated the effect of poly-(C) tract truncation in the context of initial rates of gro*wt*h of infectious virus. For this, we generated an additional infectious copy plasmid containing 29 cytosines for comparison against the C11 and C35 (*wt*) versions. These infectious copies contain the complete FMDV genome sequence including both Leader and the capsid coding region, but do not encode GFP.

RNA transcripts were made from these plasmids and transfected into cells in a CPE development assay to investigate the effect of these truncations on virus rescue, CPE development and cell-to-cell spread (Fig 5D). However, no clear differences were observed, and although the C11 transcript appeared to induce slightly less CPE than the other two transcripts initially, this was not statistically significant. The impaired encapsidation observed in the *trans*-encapsidation assay was therefore not reflected in the recovery of infectious virus.

These infectious copy RNA transcripts were subsequently used as capsid-donors in a 'reverse' *trans*-encapsidation assays, using the *wt* ΔLbdcap (replicon lacking functional capsid) as the genome-donor in each case. In this scenario, it was expected that the more competitive the capsid-donor RNA is at packaging, the fewer the resources available for replicon RNA packaging, resulting in reduced *trans*-encapsidation efficiency. In support of this prediction, the *trans*-encapsidation efficiency of the genome donor replicon was reduced when using capsid donor infectious copy transcripts with longer (C29, C35) poly-(C) tracts, and replicon *trans*-encapsidation increased when using the capsid donor infectious copy transcript with a shorter (C11) poly-(C) tract. Although there was also a slight increase in replicon *trans*-encapsidation when using the C29 capsid donor transcript over the *wt* capsid donor transcript, this was not statistically significant (Fig 5E). This was consistent with the results described earlier with the standard *trans*-encapsidation assay using the poly-(C) tract-modified replicons.

In summary, these findings demonstrate that longer poly-(C) tracts confer better packaging efficiency. This represents the first clear role that has been identified for this feature of the FMDV genome in the virus lifecycle.

## The poly-(C) tract extends rapidly due to selection pressure from encapsidation

The FMDV poly-(C) tract is typically >100 cytosines in length in *wt* virus genomes [11], which is considerably longer than the poly-(C) tracts present in the RNA transcripts used in this study. It is only possible to consistently maintain a poly-(C) tract length of around 35 nt in the infectious copy plasmid, which is why the poly-(C) tract lengths of our constructs are considerably shorter than the poly-(C) tracts found in *wt* virus genomes. However, providing they are greater than a minimal length, poly-(C) tracts shorter than *wt* have not been reported to effect replication, translation, virus viability or genome packaging. Even when virus was recovered from the C11 transcript alongside the C35 transcript, there was no noticeable difference in terms of the speed of rescue (Fig 5D). However, short poly-(C) tracts have been reported to revert to a *wt* length of 75–100 nt. during virus growth, thus obscuring effects the truncations

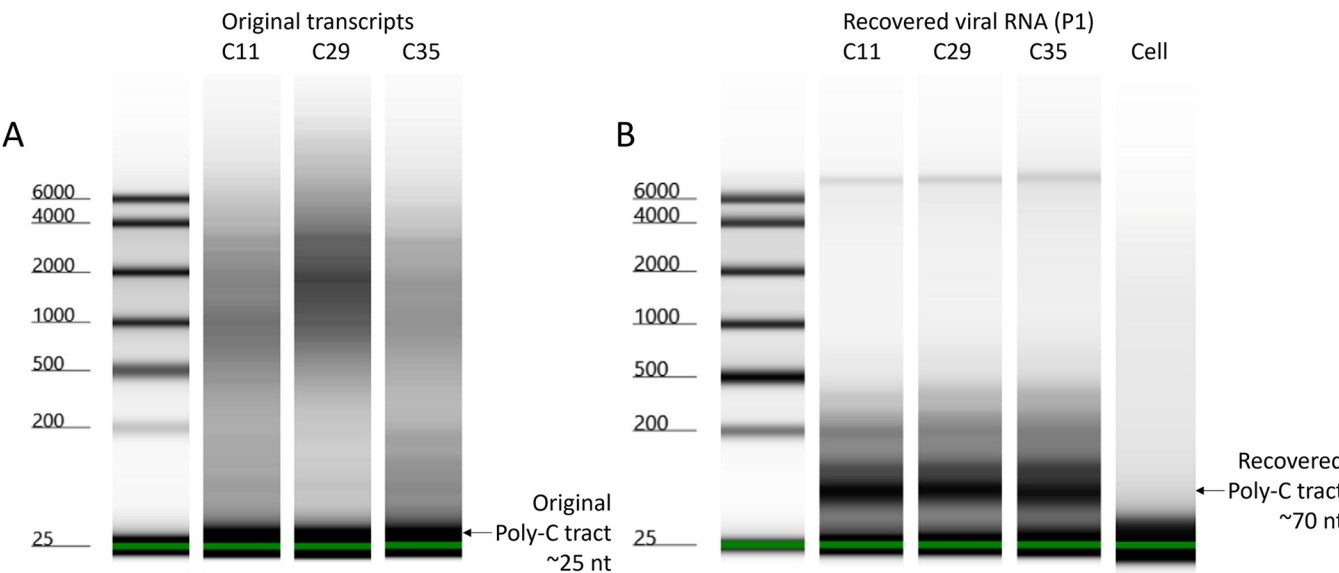

**Fig 6. Truncated poly-(C) tracts extend to near *wt* length during the first passage.** Electrophoresis analysis using an Agilent 4200 Tapestation of RNase T1 digests of; (A) RNA samples from the *in vitro* transcription reactions and (B) the corresponding extracted RNAs after transfecting the transcripts and passaging the recovered viruses. Lane 1 in each contains the RNA ladder; Lane 2 the C11 sample, Lane 3 the C29 sample; and Lane 4 the *wt* C35 sample. For the recovered viral RNA, a cell only control is shown in Lane 5. The lower marker is at 25 nt. The fragments containing the poly-(C) tracts are indicated by arrows on the right-hand side.

may have on the virus [15, 16]. To determine if this extension of the poly-(C) tract is rapid enough to explain the apparent *wt* viability of transcripts with severely truncated poly-(C) tracts, we compared the lengths of the poly-(C) tracts in RNA transcripts derived from the three infectious constructs, C11, C29, C35 (*wt*), with those present in the recovered viruses. Viruses were recovered by transfection of cells with transcript RNA, followed by a single passage on new cells until development of complete CPE (approximately 24 hours). This ensured that the majority of the viral RNA taken forward for analysis was derived from virions produced during the initial transfection, thus having passed through the packaging bottleneck at least once.

Transcripts or recovered RNA were digested with RNase T1 and resulting RNA fragments were analysed by automated electrophoresis to assess the lengths of the RNase T1 resistant poly-(C) tracts in each case [40]. A broad peak at ~25 nt was seen in digests of the transcripts derived from the original modified plasmids (Fig 6A) whereas a major and similarly broad peak at ~70 nt was seen in digests of the RNAs from virus recovered after a single passage following transfection (Fig 6B), as expected for a newly recovered poly-(C) tract [16]. The broadness of these peaks indicated that the length of the poly-(C) tract within each population was varied, but clustered around the stated lengths, which was expected and consistent with the presence of different quasispecies. While this prevents us from determining an exact poly-(C) tract length for each construct, the resolution is still sufficient to show the increase from the original transcripts to the recovered virus. This suggests that the rapid restoration of the poly-(C) tract length obscures differences between the three viruses in the CPE development assay and hinders the study of encapsidation.

## At least one pseudoknot is required for encapsidation

Our previous work on the PK region showed that PK deletions did not prevent RNA replication, but appeared to be detrimental to virus assembly, as inferred from the lack of virus

recovery [17]. To investigate the hypothesised loss of packaging associated with PK deletions, we used modified replicons in the *trans*-encapsidation assay.

These replicons encompassed distinct deletions within the PK region, each within the ΔP1 GFP replicon backbone, i.e.: ΔPK34, with PKs 3 and 4 deleted; ΔPK234, with PKs 2, 3 and 4 deleted; and ΔPK1234, with all four PKs deleted (Fig 1). Due to cloning difficulties arising from the proximity of the PK region to the poly-(C) tract, the ΔPK1234 replicon contained a poly-(C) tract of only 11 cytosines as opposed to the approximately 35 cytosines present in the poly-(C) tracts of the other replicon plasmids. We verified that the stability of the corresponding RNA transcripts in cells were comparable using RT-qPCR analysis of viral RNA at six hours post-transfection in the presence of GuHCl to block RNA replication (S2 Fig).

To differentiate effects on replication of the truncated poly-(C) tract from the effects of the complete PK region deletion, we previously made a C11 replicon with all four PKs present but containing a truncated poly-(C) tract with 11 cytosines. This C11 replicon was used here in a *trans*-encapsidation experiment to clarify whether defects in packaging were due to the deletion of the PK region, the truncation of the poly-(C) tract or a combination of both.

In this *trans*-encapsidation experiment (as in Fig 2), the replication kinetics of the C11, ΔPK34 and ΔPK234 replicons were similar in terms of GFP expression in the first round of the assay. This was assessed both from the GFP object count (Fig 7A), which represents the

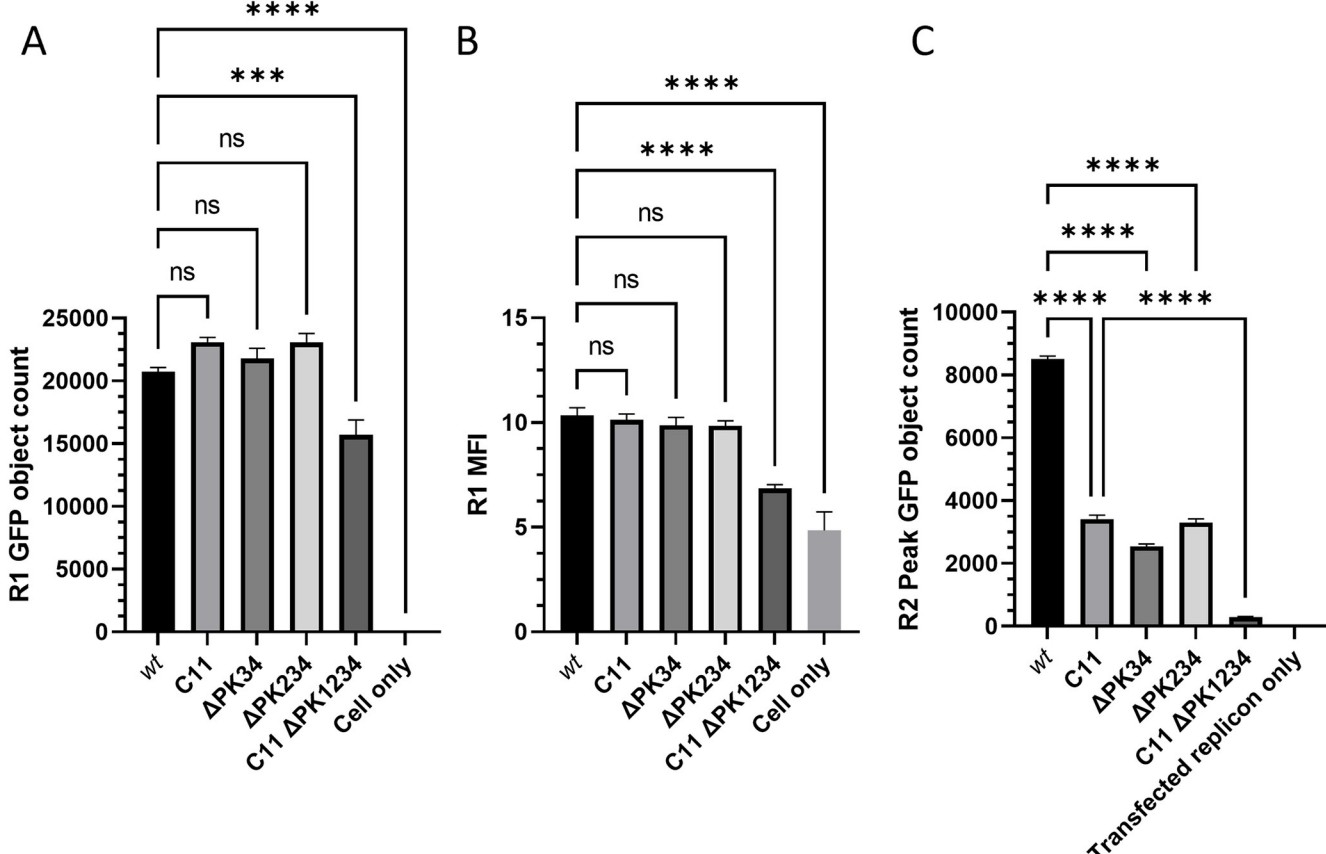

**Fig 7. Replicons lacking the pseudoknots and poly-(C) tract are not *trans*-encapsidated.** Replicons containing deletions in the PK region and/or poly-(C) tract were used in the *trans*-encapsidation assay. The (A) GFP object count and (B) green MFI of the first round are shown, along with the (C) GFP object count of the second round. The data shown represent the mean from triplicate wells, and the error bars represent the SEM from 12 images. Significance is shown compared to the *wt* GFP replicon using a one-way ANOVA (**** p < 0.0001).

number of cells in which the GFP replicon is replicating based on the parameters defined in the analysis, and from the MFI (Fig 7B). This indicates that neither RNA replication nor translation were impaired. This allowed direct comparison of these constructs in the second round of the assay which assesses encapsidation efficiency. However, the C11 ΔPK1234 replicon produced significantly fewer and less intense GFP foci (Fig 7A and 7B), consistent with our previous observations [17]. When tested by RT-qPCR, it was observed that the Ct value at the point of harvest for the C11 ΔPK1234 replicon was greater than the values of the other replicons, indicating that RNA replication was indeed impaired (S2 Fig). Despite this reduction in replication kinetics, it was still tested in the second round of the assay, but considerations were made when drawing conclusions to account for the effects that the reduced RNA replication of the C11 ΔPK1234 replicon would have on encapsidation.

In the second round, it was apparent that all three ΔPK mutant transcripts had been packaged less efficiently *in trans* compared to their parent replicons (*wt* GFP replicon compared to the ΔPK34 and ΔPK234 replicons and C11 GFP replicon compared to the ΔPK1234 replicon) (Fig 7C). *Trans*-encapsidation efficiency for the C11 replicon was also reduced, consistent with the previous findings in this study (Fig 4).

Most notably, packaging of the C11 ΔPK1234 transcripts was severely reduced with a 12-fold reduction in the number of GFP positive cells per well compared to the C11 replicon. In contrast, the peak levels for the ΔPK234 and ΔPK34 replicons were only reduced by 2.5- and 3.3-fold respectively compared to the *wt* replicon (Fig 7C). Although the C11 ΔPK1234 replicates less well than the other transcripts (as described above and in Fig 7A and 7B), this reduction was thought to be insufficient to account for the near complete loss of encapsidation and the complete loss of virus recovery seen in our previous study [17]. A caveat for the interpretation of these results is that the reduced level of replication seen with the C11 ΔPK1234 construct suggests that it may produce less replicon RNA for packaging compared to the other replicons.

Therefore, from these data alone, it was uncertain exactly why the C11 ΔPK1234 replicon was packaged so poorly. It may be because PK1 (or at least one PK if not specifically PK1), is essential; or it may be a compound outcome resulting from the effects of two mutations that are detrimental for packaging in addition to the poor replication of the replicon. Although a PK appears to be required for packaging, it is not clear whether PK1 specifically is necessary for efficient *trans*-encapsidation or if any of the PKs would suffice.

## PK1 is required for *wt* levels of encapsidation

To test if the PK1 deletion alone was attributable for the loss of *trans*-encapsidation efficiency, or if it was the result of multiple detrimental mutations, individual PK deletions were introduced into the replicon and tested in the assay. All four PKs have extremely similar sequences, suggesting that each may be able to functionally replace PK1 to facilitate packaging. PK1 and PK2 are particularly similar, differing only within a region of 11 nt at the 5' end (Fig 8A). Therefore, PK1 and PK2 were deleted individually in the ΔLbdcap GFP replicon (Fig 1) and tested in the *trans*-encapsidation assay (Fig 2). Due to limitations in the accuracy of the DNA synthesis, the poly-(C) tract length of the two constructs differed by one cytosine; the ΔPK1 replicon had 39 nt (C39 ΔPK1 GFP ΔLbdcap) and the ΔPK2 replicon had 40 nt (C40 ΔPK2 GFP ΔLbdcap) in this tract according to the manufacturer. It was not possible to Sanger sequence through the poly-(C) tract, so we instead verified the poly-C tract lengths by analysing the lengths of DNA fragments from the plasmid using electrophoresis (Fig 8F), and found that the C39 ΔPK1 GFP ΔLbdcap and C40 ΔPK2 GFP Δlbdcap replicons were both 40 nt shorter than the *wt*, consistent with a 4 (ΔPK1) and 5 (ΔPK2) nt increase in the poly-(C) tract

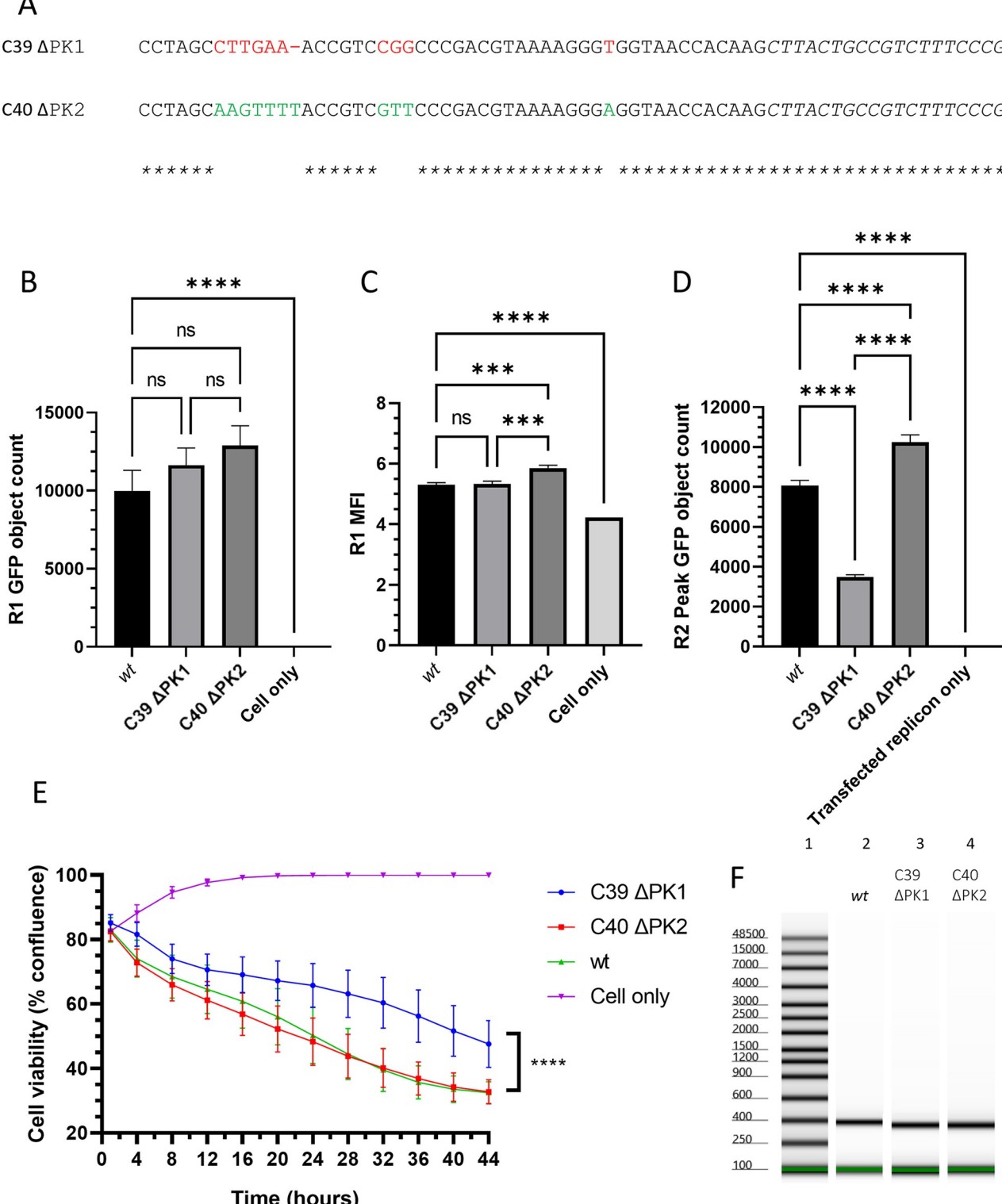

**Fig 8. Replicons lacking PK1 have reduced *trans*-encapsidation efficiencies.** (A) An alignment of the 5'-end of the PK region is shown for the ΔPK1 and ΔPK2 deletion mutants, using red to denote nucleotides unique to PK2 (in ΔPK1, top) or green to denote those unique to PK1 (in ΔPK2, bottom). Replicons containing these mutants were tested in the *trans*-encapsidation assay, and the (B) first round GFP object count, (C) first round green MFI and (D) second round GFP object count are reported. (E) CPE development assay using transfected virus transcripts containing the deletions. The data shown represent the mean from triplicate wells, and the error bars represent the SEM from 12 images. Significance is shown compared to the *wt*

GFP replicon using a one-way ANOVA (B-D) or by using a Wilcoxin test between the samples and the *wt* transcript (E) (**** p < 0.0001). (F) The plasmids were digested using *Xba*I and *Nhe*I, the fragments of interest gel purified and analysed by electrophoresis on a Tapestation. The ladder is shown in Lane 1, the C35 *wt* replicon plasmid in Lane 2, the C39 ΔPK1 plasmid in Lane 3 and the C40 ΔPK2 plasmid in Lane 4. A single experiment is shown, representative of three experimental repeats. In this experiment, the fragment from the C35 *wt* plasmid was estimated to be 395 nt (expected size 351 nt) and the C39 ΔPK1 and C40 ΔPK2 fragments were estimated as 355 and 357 nt respectively (expected sizes 311 nt and 313 nt). While the estimated lengths are different from the predicted lengths, the differences between the *wt* and the ΔPK fragments were 40 nt, as predicted from the sequences.

and a 44 (ΔPK1) and 43 (ΔPK2) nt reduction due to the PK deletion. Both poly-(C) tracts were greater in length compared to the estimated 35 cytosines in the *wt* ΔLbdcap replicon poly-(C) tract. These constructs were deemed to be sufficiently close in poly-(C) tract length to enable reliable comparisons of their *trans*-encapsidation efficiencies.

In the first round of the *trans*-encapsidation assay, deletion of either PK1 or PK2 did not significantly diminish the levels of replication of each replicon; although both had higher GFP object counts compared to the *wt* replicon, these differences were not significant (Fig 8B). Additionally, the C39 ΔPK1 ΔLbdcap GFP replicon expressed GFP to a similar level of intensity as the *wt* replicon while the C40 ΔPK2 ΔLbdcap GFP replicon produced significantly more intense GFP fluorescence than either of the other two (Fig 8C). The similarity of the replication of these replicons made direct comparisons in the second round of the *trans*-encapsidation assay possible.

In the second round, the peak GFP count for both replicons was reached at equivalent times but with significantly different values; ΔPK1 was encapsidated less well than the *wt* replicon (at 43%), while the ΔPK2 replicon was encapsidated significantly more efficiently (at 127%) than the *wt* replicon. However, this may be due to differences in replication ability as evidenced by the GFP signals in the first round (Fig 8D).

To reinforce the results of the *trans*-encapsidation assays by the analysis of virus viability, the individual PK deletions were also introduced into infectious copy plasmids, and RNA transcripts were then transfected into cells for the CPE development assay. The recovery of infectious virus and development of CPE after these transfections was based on the monolayer confluence; decreases in confluence represent progression of CPE through cell rounding and cell lysis/death, which both result in decreases in the area occupied by infected cells. CPE development was thus used to determine relative rates of progression of infection through cell monolayers, following transfection.

Following transfection, the ΔPK1 virus spread through the cells and caused CPE more slowly than either the *wt* or the ΔPK2 virus, which were comparable to each other (Fig 8E). This delay in virus spread could be attributed to the defects in encapsidation observed in the equivalent constructs in the *trans*-encapsidation assay (Fig 8D), with the reduced rates of encapsidation delaying the production of progeny virions.

## Discussion

In earlier published work, *trans*-encapsidation in FMDV was reported to be inefficient, especially when compared to PV [34]. It was speculated that this could be due to either a preference for FMDV genomes to be packaged into capsid proteins provided *in cis*, or a preference for full sized genomes to be packaged over the smaller replicons, but neither of these hypotheses was verified.

The current study indicates that a major issue with the previously published FMDV *trans*-encapsidation assays was that packaging of the replicon RNA was outcompeted by the *wt* helper virus RNA. This is in part due to the lack of the capsid coding region in the replicon, and consequent absence of PSs present in that region [35]. However, the earlier *trans*-

encapsidation studies with PV replicons showed relatively efficient *trans*-encapsidation despite equivalent deletion of the PV capsid region [25]. Enteroviruses likewise contain PSs dispersed across the entire genome, including in the capsid coding region [25], suggesting that the deletion of the capsid coding region in the FMDV replicon is unlikely to be the sole reason for inefficient *trans*-encapsidation.

This inefficiency could be overcome in part by reducing the competitiveness of the capsid-donor. Crucially, by competing with virus RNA transcripts modified to contain shorter poly-(C) tracts than those in the replicon, the replicon was encapsidated with much greater efficiency. Considering that previous iterations of the assay have used *wt* virus as the capsid-donor, and that the *wt* virus typically contains a poly-(C) tract with over 100 cytosines [11], it is likely that the reduced *trans*-encapsidation efficiency previously observed can be directly correlated to the length of the poly-(C) tract in the capsid-donor used. PV does not contain a poly-(C) tract and so this feature cannot explain differences in *trans*-encapsidation of PV and FMDV replicons. To counter this problem in FMDV, we used a transcript with a truncated poly-(C) tract as the capsid-donor. This reduced the competition faced by the FMDV replicon, making the *trans*-encapsidation assay a sensitive, specific and reproducible method for determining the effects of various mutations on the packaging of FMDV genomes.

The *trans*-encapsidation assay allowed us to confirm that PSs play an important role in FMDV encapsidation. Not all PSs in the genome have equal effects, however. The PSs tested functionally by Logan et *al.* [35], namely PS2, PS3 and PS4, cumulatively resulted in a drop in virus viability compared to the *wt* virus, but the viruses were still able to package and propagate. Similarly, re-introducing PS3-5 to the GFP replicon by using a ΔLb GFP virus as the recipient resulted in an increase in encapsidation efficiency compared to the ΔP1 GFP replicon, although the ΔP1 GFP replicon was still packaged, indicating that these PSs alone are clearly not essential.

In contrast, the PS(s) in the PK region appear to be far more important for packaging compared to those present in the capsid coding region. Deleting the PK region almost abolished replicon *trans*-encapsidation, and while the deletions in C11 ΔPK1234 also affected the replicon replication kinetics, this is not sufficient to explain the reduction in *trans*-encapsidation.

Deleting only parts of the PK region also severely hindered encapsidation, but without affecting replication or translation. PK1 specifically appeared to be of particular importance, as deleting it severely reduced encapsidation whereas the equivalent deletion of PK2 seemingly increased encapsidation, although this may be attributable to either the increased length of the poly-(C) tract in this transcript compared (C40) to the *wt* replicon (C35) or to the increase in replicon replication ability observed from the GFP MFI in the first round of the assay. PK1 also corresponds to the putative PS with the greatest constraint that Logan et *al.* [35] identified, supporting the hypothesis that the PS in this position is particularly important for FMDV encapsidation. However, a considerable amount of encapsidation still occurred when PK1 alone was deleted, indicating that PK1 itself is not essential. An explanation for this can be seen from the alignment of PK1 and PK2 (Fig 8A); aside from a stretch of nucleotides near the 5' end, the sequences are almost identical and PK2 is thus likely able to adopt the role of the PS. This similarity does include PK3 and PK4, although the sequences do diverge slightly more for the latter two PKs. When larger portions of the PK region were deleted, in ΔPK234 and ΔPK34, *trans*-encapsidation was similarly reduced. This reduction could be attributed to either "off-site" effects disrupting the PS in PK1 or to the deletion of active PSs. If the latter, the redundancy of the PKs could be since multiple PSs may act in concert to strengthen the PS interaction, with the 5'-most PS (PK1) being particularly important.

Due to the importance of the PK region and the PS(s) that it contains for encapsidation, we hypothesise that the influence the poly-(C) tract has on encapsidation is due to its proximity to

PK1 and PS1. Rather than being a PS itself, the poly-(C) tract may instead act as a spacer between the PK region and the highly structured S-fragment, allowing the PS to fold independently of upstream components. A minimal number of cytosines may be necessary to achieve this, with an optimal number likely to be about 100 as seen in the *wt* virus [11]. This requirement could explain why viruses with truncated poly-(C) tracts rapidly recover their poly-(C) tracts and why all strains of FMDV contain this unusual feature, as they are selected for during the encapsidation bottleneck.

The role for the poly-(C) tract in encapsidation has likely remained undiscovered for two reasons. Firstly, the poly-(C) tract can rapidly recover from truncations. Truncations leaving at least six nucleotides in the poly-(C) tract have been reported to revert to *wt* length [16], and here we have demonstrated the same for an eleven nucleotide poly-(C) tract after one passage. This makes investigations into its role difficult since the length is so variable. Constructs must therefore be assessed directly following transfection instead of using recovered virus, because the process of rescuing virus allows the poly-(C) tract to extend and potentially nullify the effects of truncations within the original plasmid constructs. Secondly, results may have previously been obscured by the lack of competition from more packaging-competent constructs. Here, we have shown that although genomes containing longer poly-(C) tracts are packaged preferentially, the rates of virus recovery from constructs with poly-(C) tracts of different lengths were indistinguishable in the absence of competition.

It should be noted that a virus containing a poly-(C) tract with only two nucleotides (C2) has been shown to infect mice and to grow in cell culture, but to significantly lower titres than *wt* virus [16], possibly due to compromised encapsidation. Virus recovered from this C2 construct often had a 42 nt deletion, corresponding to PK1, which we have highlighted as an important PS [35]. We have shown that constructs lacking only PK1 are still able to be encapsidated *in trans*, albeit inefficiently, suggesting that similar sequences in the other PKs are able to act as substitutes albeit at reduced efficiency. However, this reduced efficiency might only be true when PK1 is in the presence of a poly-(C) tract that can help it fold into the correct PS confirmation. It is therefore possible that when combined with a severely truncated poly-(C) tract, which prevents PK1 from folding correctly, deletion of PK1 may instead be beneficial and selected for during encapsidation.

A potential model for packaging based on these results proposes that the PK region can adopt two alternative conformations; the first being the typical PK region, comprising four pseudoknots, and the second being a packaging competent conformation in which a PS is exposed and able to bind to a capsid precursor and initiate encapsidation (Fig 9). This would enable the RNA to follow one of two pathways depending on the conformation adopted; a packaging focussed pathway, and a replication focussed pathway. This transition between two RNA conformations to control separate aspects of the virus lifecycle is a mechanism which has been observed in other RNA virus families [41–43].

In this proposed model, as a newly replicated positive strand RNA molecule emerges from the replication complex, the RNA of the PK1/PS1 region initially adopts a transient conformation displaying the 'primary' PS, PS1. In this nascent state the PS1 binds to a pentamer subunit to stabilise PS1 in the packaging conformation and initiate encapsidation. This RNA-pentamer complex may then act as the nucleation site for capsid assembly with the emerging replicated RNA engaging additional pentamers through 'secondary' PSs and arranging them strategically within the assembling capsid. This may assist folding of the FMDV genome into a compact conformation with a small hydrodynamic radius based on the PS interactions with the nearby pentamers. The resulting particle is converted to infectious virus by maturation cleavage of VP0 into VP2 and VP4. Alternatively, if the RNA emerging from the replication complex fails to interact with a capsid precursor, the transient PS conformation collapses into a more

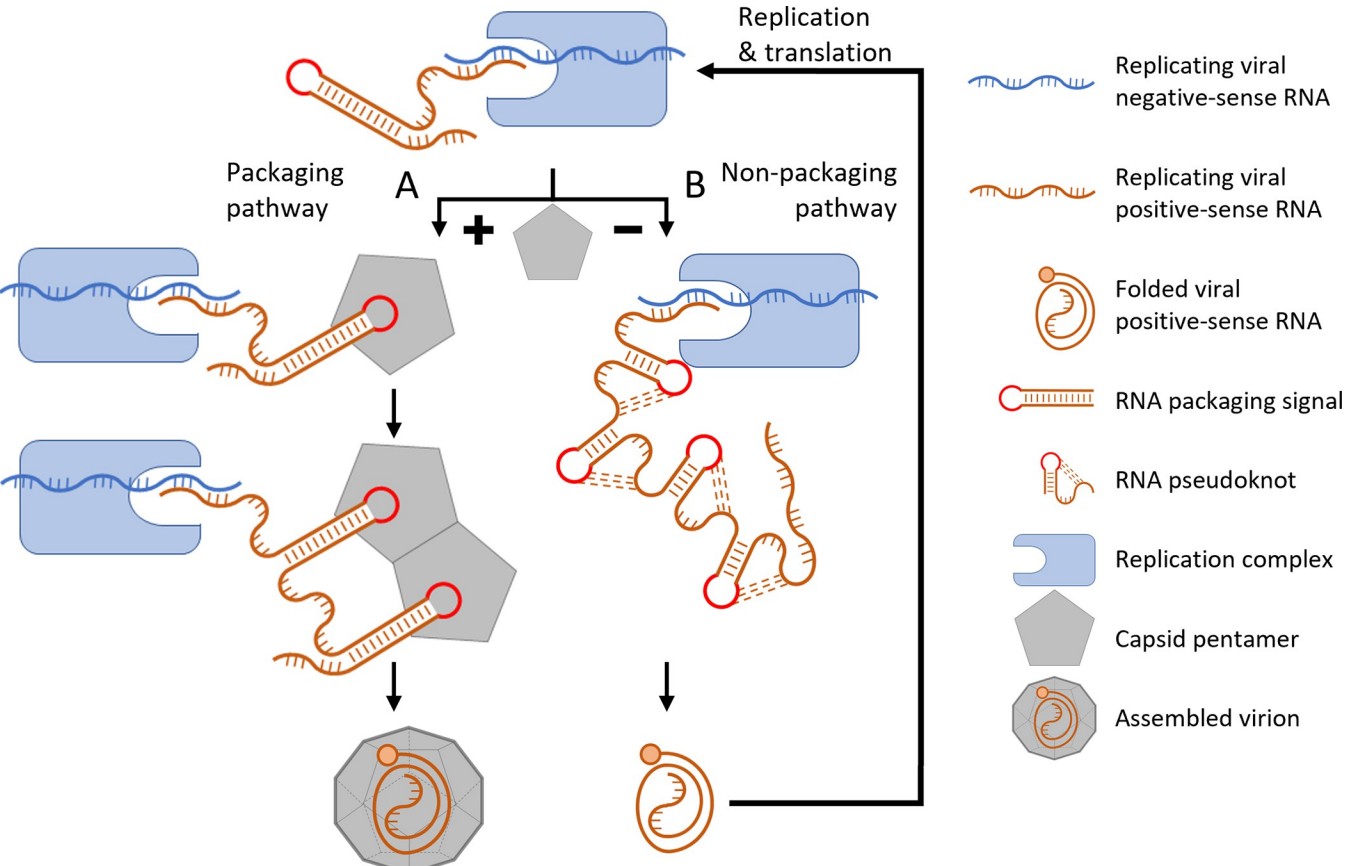

**Fig 9. Model for capsid assembly.** Negative-sense RNA (blue) acts as the template for the formation of corresponding positive-sense strand RNA molecules (brown), and PS1 forms a stem-loop as the RNA emerges from the replication complex. (A) If nascent PS1 encounters a pentamer, the interaction stabilises the RNA in the stem-loop conformation. Subsequent packaging signals then form as the RNA continues to emerge from the replication complex and these interact with additional pentamers to form the 'nucleus' of capsid assembly. Once completed, the genome is completely encapsidated within the new virion. (B) If no pentamers are present to stabilise PS1, the RNA stem-loop collapses into the pseudoknot conformation and the completed positive-sense strand is used for translation and/or further RNA replication.

energetically favourable PK conformation, and the RNA serves as a template for further replication and/or for translation.

In this model, we have divided the PSs into two categories–a singular primary PS, which we hypothesise to be essential for genome encapsidation and to act as the encapsidation primer; and secondary PSs, which individually are not essential, but each contributes to the cumulative effect on encapsidation. This model, involving the interplay between PSs and pentamer sub-unit concentration, may explain how picornavirus encapsidation is highly specific and why it only involves actively replicating RNA [44]. It also provides a mechanism for optimising the timing of the transition from replication and translation to packaging.

Furthermore, this model is compatible with encapsidation models proposed previously. In PV, for example, earlier work has shown that protein-protein interactions are key for encapsidation [4,33], with less importance attributed to RNA PSs [45]. Putative RNA PSs have, however, since been identified throughout the PV genome [25], suggesting that PV encapsidation can make use of both protein-protein interactions and RNA-protein interactions. Even when large sections of the genome have been recoded, due to the comparatively lower individual contributions secondary PSs provide, the presence of any remaining secondary PSs may be

sufficient to enable encapsidation, with an element of redundancy in the secondary PS system. The structure of PSs is also thought to be fairly simple, consisting of a small stem-loop with a purine stack at the distal point of the loop, so it is also feasible that equivalent stem-loops were added during the recoding which helped facilitate encapsidation.

We propose that both protein-protein and RNA-protein interactions could act in concert in our model, where the protein-protein interactions facilitate the recruitment of capsid precursors to the replication complex in picornaviruses, providing opportunities for the RNA and the capsid precursors to interact. Following this localisation, a primary PS initiates encapsidation and secondary PSs dispersed throughout the genome facilitate the RNA encapsidation process, as suggested by our model.

## Supporting information

**S1 Fig.** (A) Replicon GFP expression and trans-encapsidation efficiency in the presence of an RNA replication inhibitor. The ΔP1 and ΔLbdcap GFP replicons were transfected into cells with (green and orange lines) and without (purple and black lines) the presence of GuHCl, a replication inhibitor. The green MFI was read over time using the Incucyte S3 Live cell imager to measure GFP expression levels from both replicating RNA and from input-only translation. The data shown represent the mean from triplicate wells, and the error bars represent the SEM from 12 images. (B) Replicon GFP expression and trans-encapsidation efficiency in the presence of an RNA replication inhibitor. The readout of the second round of the trans-encapsidation assay, measured by counting the number of GFP foci, when the first round was carried out in the presence or absence of GuHCl. The data shown represent the mean from triplicate wells, and the error bars represent the SEM from 12 images. (C, D) Mean and Standard Error data for S1 Fig A and B. (E) Representative images analysed using the Incucyte software to obtain the: MFI data at the point of harvest for: (i-vi) S1 Fig A ΔP1 and ΔLbdcap GFP replicons, in the presence and absence of GuHCl, and the cell only and capsid-donor only controls; and (vii-xii) peak GFP object count data for S1 Fig B ΔP1 and ΔLbdcap GFP replicons, following a transfection in the first round of the assay in the presence or absence of GuHCl, and the cell only and capsid-donor only controls.
(PDF)

**S2 Fig.** (A) Relative stability and replication ability of transfected replicon RNA in cells. Replicon RNA transcripts were transfected into cells, washed after 15 minutes and harvested after either 15 minutes or 6 hours post-transfection (equivalent to the harvest timepoint for the trans-encapsidation assay) in the presence or absence of 3 mM GuHCl. After extracting the RNA, relative RNA yields were assessed by RT-qPCR based on the Ct values. The experiment was performed in triplicate, with the means and standard errors shown here. No significant differences were seen between the constructs with PK deletions and the equivalent sample for the wt replicon ($P < 0.005$), nor were marked differences seen in the presence or absence of GuHCl for each construct. The dotted line represents the cut-off point. (B) RNA abundance values, including the Mean and Standard Error data, for S1A. Values were calculated based on the Ct values (assuming 100% amplification efficiency) and multiplied by $10^{12}$ to improve the scale for ease of interpretation.
(PDF)

**S3 Fig.** (A-C) Mean and Standard Error data for Fig 3A–3C respectively. (D) Representative images analysed using the Incucyte software to obtain the: GFP object count and MFI data at the point of harvest for Fig 3A and 3B (i) ΔP1 GFP replicon, (ii) ΔLbdcap GFP replicon, (iii) Cell only and (iv) Capsid-donor only; and peak GFP object count data for Fig 3C (v) ΔP1 GFP

replicon, (vi) ΔLbdcap GFP replicon and (vii) transfected replicon only.
(PDF)

**S4 Fig.** (A-E) Mean and Standard Error data for Fig 5A–5E respectively. (F) Representative images analysed using the Incucyte software to obtain the: (i-iv) GFP object count and MFI data at the point of harvest for Fig 5A and 5B ΔP1 C11, C29 and C35 GFP replicons and cell only control; (v-viii) peak GFP object count data for Fig 5C ΔP1 C11, C29 and C35 GFP replicons and transfected replicon only control; (ix-xx) monolayer confluency to measure CPE for the C11, C29 and C35 ICs and cell only control at 0, 12 and 24 hrs for Fig 5D; (xxi-xxiv) and peak GFP object count data for Fig 5C ΔP1 C11, C29 and C35 GFP replicons and transfected replicon only control for Fig 5E.
(PDF)

**S5 Fig.** (A-C) Mean and Standard Error data for Fig 7A-C respectively. (D) Representative images analysed using the Incucyte software to obtain the: (i-vi) GFP object count and MFI data at the point of harvest for Fig 7A and Fig 7B ΔP1 wt, C11, ΔPK34, ΔPK234 and C11 ΔPK1234 GFP replicons and cell only control; and peak GFP object count data for Fig 7C (vii-xii) ΔP1 wt, C11, ΔPK34, ΔPK234 and C11 ΔPK1234 GFP replicons and transfected replicon only control.
(PDF)

**S6 Fig.** (A-D) Mean and Standard Error data for Fig 8B–8E. (E) Representative images analysed using the Incucyte software to obtain the: (i-iv) GFP object count and MFI data at the point of harvest for Fig 8B and Fig 8C wt, C39 ΔPK1 and C40 ΔPK2 ΔLbdcap GFP replicons and cell only control; peak GFP object count data for Fig 8D (v-viii) wt, C39 ΔPK1 and C40 ΔPK2 ΔLbdcap GFP replicons and transfected replicon only control; and (ix-xx) monolayer confluency for CPE measurement for wt, C39 ΔPK1 and C40 ΔPK2 ICs and cell only control at 0, 12 and 24 hrs.
(PDF)

## Acknowledgments

The authors would also like to acknowledge Prof. Stephen Curry for his support as a supervisor to CN during his PhD.

## Author Contributions

**Conceptualization:** Chris Neil, Graham J. Belsham, Tobias J. Tuthill.

**Data curation:** Chris Neil.

**Formal analysis:** Chris Neil.

**Funding acquisition:** Graham J. Belsham, Tobias J. Tuthill.

**Investigation:** Chris Neil.

**Methodology:** Chris Neil, Joseph Newman, Tobias J. Tuthill.

**Project administration:** Chris Neil, Joseph Newman, Tobias J. Tuthill.

**Supervision:** Joseph Newman, Graham J. Belsham, Tobias J. Tuthill.

**Validation:** Chris Neil.

**Visualization:** Chris Neil.

**Writing – original draft:** Chris Neil, Tobias J. Tuthill.

**Writing – review & editing:** Chris Neil, Joseph Newman, Nicola J. Stonehouse, David J. Rowlands, Graham J. Belsham, Tobias J. Tuthill.

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
