## [Decision Letter · Decision Letter 0]

24 Jul 2024

Dear Dr Neil,

Thank you very much for submitting your manuscript "The pseudoknot region and poly-(C) tract comprise an essential RNA packaging signal for assembly of foot-and-mouth disease virus" for consideration at PLOS Pathogens. As with all papers reviewed by the journal, your manuscript was reviewed by members of the editorial board and by several independent reviewers. In light of the reviews (below this email), we would like to invite the resubmission of a significantly-revised version that takes into account the reviewers' comments.

The reviewers have raised important concerns which should be addressed.

We cannot make any decision about publication until we have seen the revised manuscript and your response to the reviewers' comments. Your revised manuscript is also likely to be sent to reviewers for further evaluation.

Sincerely,

WEIDONG XIAO

Guest Editor

PLOS Pathogens

Ashley St. John

Section Editor

PLOS Pathogens

Michael Malim

Editor-in-Chief

PLOS Pathogens

orcid.org/0000-0002-7699-2064

The reviewers have raised important concerns which should be addressed.

Reviewer's Responses to Questions

**Part I - Summary**

Reviewer #1: In the manuscript submitted by Chris Neil and Joseph Newman et al., the authors use two types of replicon: ΔP1 GFP replicon and ΔLbdcap GFP replicon to investigate FMDV RNA packaging through a novel trans-encapsidation assay, and reveal the essential role of packaging signals in FMDV RNA packaging, specifically those in the poly-(C) tract and pseudoknot (PK) region, where the length of the poly-(C) tract and PK1 influence the efficiency of RNA encapsidation. Based on this, they propose a simple model for FMDV RNA packaging, which involves a transition from genome replication to genome packaging and is controlled by packaging signals. These found could provide scientific guidance for the research and development of new antiviral strategies targeting FMDV and other picornaviruses.

Reviewer #2: In this well-written manuscript, the authors describe a well-crafted study aimed at the long-standing question of how picornaviruses, positive-sense RNA viruses, package their genomic RNAs into virion capsids. Unlike other viruses, e.g., retroviruses, there has been no description of a single RNA sequence or structural element that directs newly-synthesized viral RNAs into newly-formed capsid structures. In the present study, that authors have chosen to address this question using FMDV as their model picornavirus, a virus that is known to infect cloven hooved animals and cause major outbreaks. The authors showed that specific RNA sequence/structure elements [including pseudoknots and the poly(C) tract in the 5’ UTR] dispersed throughout the genome play important roles in RNA packaging for FMDV. Using a clever replicon rescue, trans-encapsidation/RNA packaging assay, they identified several packaging sequences and showed that the length of the poly(C) tract impacts encapsidation, with longer tracts producing higher levels of packaging efficiency. In these studies, the authors were particularly careful to eliminate other potential variables/contributors to solidify their conclusions, using a two round assay to their advantage. This led them to propose a very plausible model to explain their data, although the model may not fit for enteroviruses like poliovirus. With one exception (see below) the Introduction and Discussion sections were thoughtfully written.

**Part II – Major Issues: Key Experiments Required for Acceptance**

Reviewer #1: 1. The author transfected approximately 8kb of replicon RNAs such as C11, ΔPK34, ΔPK234, and C11 ΔPK1234 into BHK-21 cells. These different replicon mutants would alter the RNA structure and potentially alter the stability of the RNA in the cell. The author also needs to investigate whether the stability of these transfected RNAs in cells has changed.

2. Whether replicons such as C11, C29, Δ PK234, and C11 Δ PK1234 have the same replication ability in cells, because the stronger the replication ability of the replicon RNA, the more RNA is replicated, and the higher the probability of being packaged. Therefore, in the second round of infection, the more GFP object count is present.

3. There is no difference between GFP and MFI in Figures 5a-b, which only indicates that they are similar in translation level and does not necessarily mean that there is no difference in replicon replication ability.

4. In Figure 5E, the author assesses the trans-encapsidation efficiency of the C11, C29, and C35 (WT) genome donor replicon through using C11, C29, and C35 (WT) capsid donor infectious copy transcripts, respectively (lines 355-361). How can the author determine whether GFP comes from the capsid donor infectious copy transcripts or genome donor replicon? Is it possible that the difference in R2 Peak GFP object count is due to the difference in virus rescue between C11, C29, and C35 (WT) capped donor infection copy transcripts.

5. The author used RNase T1 to digest transcripts and recovered RNA, but the recovered viral RNA contained a large amount of cellular RNA. Therefore, the digested "recovered multi C region 70" marked by the author in Figure 6B may mostly be cellular RNA, and it is recommended that the author add a cell group as a control.

6. Based on previous research, the author found that PK does not affect virus replication, but is it possible that C39 Δ PK1 and C40 Δ PK2 affect virus translation, leading to differences in GFP object and MFI？

7. In Figure 8, C39 Δ PK1 and C40 Δ PK2 are described, while in the schematic diagram of Figure 1, C40 Δ PK1 and C39 Δ PK2 are labeled, leading to confusion in the experimental results.

8. Suggest the author to use replication deficient viruses and replicons as controls to better elucidate the role of packaging signals in virus assembly.

Reviewer #2: (No Response)

**Part III – Minor Issues: Editorial and Data Presentation Modifications**

Reviewer #1: no

Reviewer #2: Prior to having this manuscript accepted for publication, the authors should address the following questions and critiques.

1. The authors have omitted a discussion of the data provided by Song and co-workers in their 2017 PNAS paper (PMID: 28973853) describing their unsuccessful search for RNA sequences/structures required for poliovirus genome encapsidation using massive genome sequence recoding. Although a much different picornavirus than FMDV, poliovirus packaging stands in contrast to what the authors have proposed in Figure 9, and the last paragraph of the Discussion does not accurately reflect the inclusion of the data from the Song et al. study. The fact that this different genus of picornaviruses has not been shown to contain one or more RNA packaging signals (despite an exhaustive recoding analysis) that guide RNA encapsidation suggests the possibility of additional signals (RNP complexes?) playing a role in encapsidation. The Discussion of the revised manuscript should contain additional consideration of this topic.

2. Line 140: The Introduction section could use a better concluding sentence stating what the study actually showed, not just to “characterize.”

3. Lines 246 and 250: Manufacturer guidelines, although a popular cop out, is not sufficient here – please describe what was done.

4. Figure 4: Why no apparent heterogeneity in the poly(C) tract here?

5. For the experiment described at the bottom of page 17, why not use RNase treatment of the recovered virus yields following the first round transfection? This would insure no carryforward of RNAs.

6. In Figure 6A, the resolution in the gel is not high enough to make a strong conclusion. Also, the experimental details of automated electrophoresis should be expanded here or in the figure legend so the reader can actually know what was done.

7. Line 453: The authors need to independently assess the poly(C) tract length and not depend on what the manufacturer states – they are not co-authors of this manuscript.

8. For the experiments described in the last paragraph of page 21, were the complete genomic sequences determined for the RNAs of the recovered viruses? This is a key question, since there may be lesions outside of the targeted regions of the FMDV genome that contribute to the phenotypes observed by the authors.

9. Line 507: What is a “robust” method? Although it is common to (mis)use this word in scientific vernacular, it really is inappropriate here.

10. In the Discussion on pages 26 and 27 (with reference to the model in Figure 9), additional considerations should be discussed. If there are two pathways (packaging and replication) for newly synthesized RNAs, how are they localized within the cytoplasm? For the RNAs in the replication pathway to be replicated, they must first be translated. Have the authors used imaging methodologies to track the newly-synthesized RNAs to distinct locations in the cytoplasm of infected cells (e.g., membrane-bound RNA replication complexes, packaging complexes, or translating ribosomes)? Even if these studies have not yet been carried out, the subcellular fate of newly synthesized FMDV RNAs should be discussed in the manuscript.

PLOS authors have the option to publish the peer review history of their article (what does this mean?). If published, this will include your full peer review and any attached files.

Reviewer #1: No

Reviewer #2: No
---

## [Decision Letter · Decision Letter 1]

4 Dec 2024

Dear Dr Neil,

We are pleased to inform you that your manuscript 'The pseudoknot region and poly-(C) tract comprise an essential RNA packaging signal for assembly of foot-and-mouth disease virus' has been provisionally accepted for publication in PLOS Pathogens.

Best regards,

WEIDONG XIAO

Guest Editor

PLOS Pathogens

Ashley St. John

Section Editor

PLOS Pathogens

Sumita Bhaduri-McIntosh

Editor-in-Chief

PLOS Pathogens

orcid.org/0000-0003-2946-9497

Michael Malim

Editor-in-Chief

PLOS Pathogens

orcid.org/0000-0002-7699-2064

Reviewer Comments (if any, and for reference):

Reviewer's Responses to Questions

**Part I - Summary**

Reviewer #1: (No Response)

Reviewer #2: In this revised manuscript, the authors have sufficiently addressed my critiques and queries. The manuscript now appears to be appropriate for publication in PLoS Pathogens.

**Part II – Major Issues: Key Experiments Required for Acceptance**

Reviewer #1: (No Response)

Reviewer #2: N/A

**Part III – Minor Issues: Editorial and Data Presentation Modifications**

Reviewer #1: (No Response)

Reviewer #2: N/A

PLOS authors have the option to publish the peer review history of their article (what does this mean?). If published, this will include your full peer review and any attached files.

Reviewer #1: No

Reviewer #2: No

---

## [Editor Report · Acceptance letter]

13 Dec 2024

Dear Dr Neil,

We are delighted to inform you that your manuscript, "The pseudoknot region and poly-(C) tract comprise an essential RNA packaging signal for assembly of foot-and-mouth disease virus," has been formally accepted for publication in PLOS Pathogens.

Best regards,

Sumita Bhaduri-McIntosh

Editor-in-Chief

PLOS Pathogens

orcid.org/0000-0003-2946-9497

Michael Malim

Editor-in-Chief

PLOS Pathogens

orcid.org/0000-0002-7699-2064